# Analysis of Arch Bridge Condition Data to Identify Network-Wide Controls and Trends

Kristopher Campbell [1,*], Myra Lydon [2], Nicola-Ann Stevens [3] and Su Taylor [3]

1   Department for Infrastructure, Belfast BT2 8GB, UK
2   Civil Engineering, College of Science & Engineering, University of Galway, University Road, H91 TK33 Galway, Ireland; myra.lydon@universityofgalway.ie
3   School of Natural and Built Environment, Queen's University, Belfast BT7 1NN, UK; nstevens01@qub.ac.uk (N.-A.S.); s.e.taylor@qub.ac.uk (S.T.)
*   Correspondence: kristopher.campbell@infrastructure-ni.gov.uk

**Abstract:** This paper outlines an initial analysis of 20 years of data held on an electronic bridge management database for approximately 3500 arch bridges across Northern Ireland (NI) by the Department for Infrastructure. Arch bridges represent the largest group of bridge types, making up nearly 56% of the total bridge stock in NI. This initial analysis aims to identify trends that might help inform maintenance decisions in the future. Consideration of the Bridge Condition Indicator (BCI) average value for the overall arch bridge stock indicates the potential for regional variations in the overall condition and the potential for human bias in inspections. The paper presents the most prevalent structural elements and associated defects recorded in the inspections of arch bridges. This indicated a link to scour and undermining for the worst-conditioned arch bridges. An Analysis of Variance (ANOVA) analysis identified function, number of spans, and deck width as significant factors during the various deterioration stages in a bridge's lifecycle.

**Keywords:** arch; bridges; inspection; data

## 1. Introduction

The Department for Infrastructure (DfI) is a government department in Northern Ireland (NI), which encompasses a range of strategic functions, including acting as the local roads authority. DfI owns and maintains approximately 7000 bridges across Northern Ireland. These bridges vary in size, geographic location, construction type, age, function, number of spans, and span lengths. DfI holds over 20 years of historic and current asset condition data for all these bridges stored on an electronic bridge management database.

The approach taken with regard to managing the information gathered during bridge inspections in NI has evolved from a paper-based system used in the 1970s to an electronic-based system introduced in 1999. During this time, bridges were given an overall priority/state rating between 1 and 5, with state 1 indicating no work was required, state 4 indicating that work was required within 1 year, and state 5 indicating there was a danger to the public, in which case the bridge needed to be closed and work needed to be undertaken as soon as possible. In 2017, DfI reviewed the approach taken and amended the bridge management system to provide additional functionality as well as changing the method of scoring bridge conditions. The Bridge Condition Indicator (BCI) scoring system was adopted as DfI believed this would provide a better indicator for prioritising maintenance and repair work than the original system. The BCI calculates a condition score between 0 and 100, with 100 indicating a bridge with no defects, and as the condition decreases, so does the BCI. However, given DfI's uncertainty about the capabilities of the BCI scoring for prioritizing work, DfI decided to continue to give the bridges an overall priority/state rating within the notes of the inspection to provide a backup position should the BCI not function as intended.

This change in the rating system had the potential to render valuable historical data obsolete. Given the lifespans of structures, this data could be used to infer the longer-term deterioration of the bridges for nearly 20 years. An exercise to calculate BCI values retrospectively based on the inspection data for inspections carried out between 1999 and 2017, allowed this historic data to be incorporated into a separate 'augmented' dataset. Details of this process are covered in work published by the authors [1].

This paper will use both the current bridge management data set and the augmented dataset at various points. The current data set consists of information held on only the most recently completed inspection for each bridge and not superseded inspection records. This will be clearly stated within the relevant sections of this paper.

## 2. Analysis of the Current Bridge Management System Data

### 2.1. Network-Wide BCI Distribution

Understanding the general condition of the NI bridge stock is critical when establishing future monitoring and assessment programs. Having recorded details for each bridge including the location, type, dimensions, and condition of the bridges allows for the analysis of network-wide considerations. It is possible to interrogate the data in order to explore potential trends or target specific issues. For example, it is possible to plot the locations of structures that exhibit certain characteristics such as plotting all bridges with a history of scour in their inspection history or plotting all bridges with a BCI lower than 60 (Xia et al., 2022).

This paper primarily focuses on arch bridge structures, which represent approximately 56% of the total DfI bridge stock in NI. This proportion is in line with the rest of the UK, with approximately 40,000 arch bridges corresponding to between 40 and 50% of the total UK bridge stock of road, rail, and navigable waterway authorities [2].

Here, 53% of the NI arch bridges are classified as masonry arch bridges, with a further 3.4% being categorised as 'Arch Other' consisting mainly of brick and concrete arches, as presented in Table 1, and representing approximately 3500 bridges and 5000 individual arched spans. Of these structures, approximately 11% have been widened or modified with the addition/replacement of individual spans.

**Table 1.** Percentages of span construction type.

| Span Construction | Percentage [1] |
|---|---|
| Masonry Arch | 53.0 |
| Reinforced Concrete Pipe | 12.4 |
| Reinforced Concrete Slab | 10.8 |
| Concrete Box Culvert | 6.8 |
| Concrete Beam (various) | 5.7 |
| Arch Other (Brick, Concrete, Jack) | 3.4 |
| Steel (Various) | 2.8 |
| Corrugated Steel Pipe | 2.7 |
| Composite Concrete and Steel | 1.2 |
| Miscellaneous | 1.4 |

[1] Percentages rounded to 1 decimal place.

For the purposes of this paper, masonry arch and arch other structures are collectively referred to as arch bridges for the remainder of the paper.

These arch bridges often have limited or no accurate information on the age, construction details, foundation type, or load capacity. This presents a significant challenge when determining the remaining design life of these assets.

The limited accurate information relating to arches highlights this structure type as a key vulnerability within UK transport networks. To enhance the resilience of the network, there is an urgent need to establish a means of identifying the remaining life of these assets and develop meaningful long-term maintenance strategies.

DfI bridges are typically subject to periodic inspection cycles as per the requirements of the CS450 Inspection of Highway Structures document within the Design Manual for Roads and Bridges (DMRB). Bridges are subject to various types of inspection, including safety, general, principal, special and inspections for assessment. Principal inspection is a more in-depth inspection, usually within touching distance, typically carried out every 6 years. General inspections are typically carried out every two years in the intervals between successive principal inspections and are not usually to the same level of detail. In this paper, all the other inspections carried out will be grouped together as other inspections. These other inspections cover inspections outside the inspection cycle, which might be focused on specific parts of a bridge or following an incident such as damage caused by a collision. At any moment in time, the most current inspection could be either a principal, general, or other inspection, and this is reflected in this paper. Using the current dataset, as of the end of April 2022, all current inspection BCI average scores for arch bridges predominantly fell between 80 and 90, followed by those between 65 and 80 and closely followed by 90–100; see Figure 1. Less than 1% of the current inspections show a BCI average value of less than 40. Initial analysis of these bridges indicates that a number are in very poor condition and are either subject to weight restrictions or road closures, are in the process of being replaced/repaired or have been abandoned.

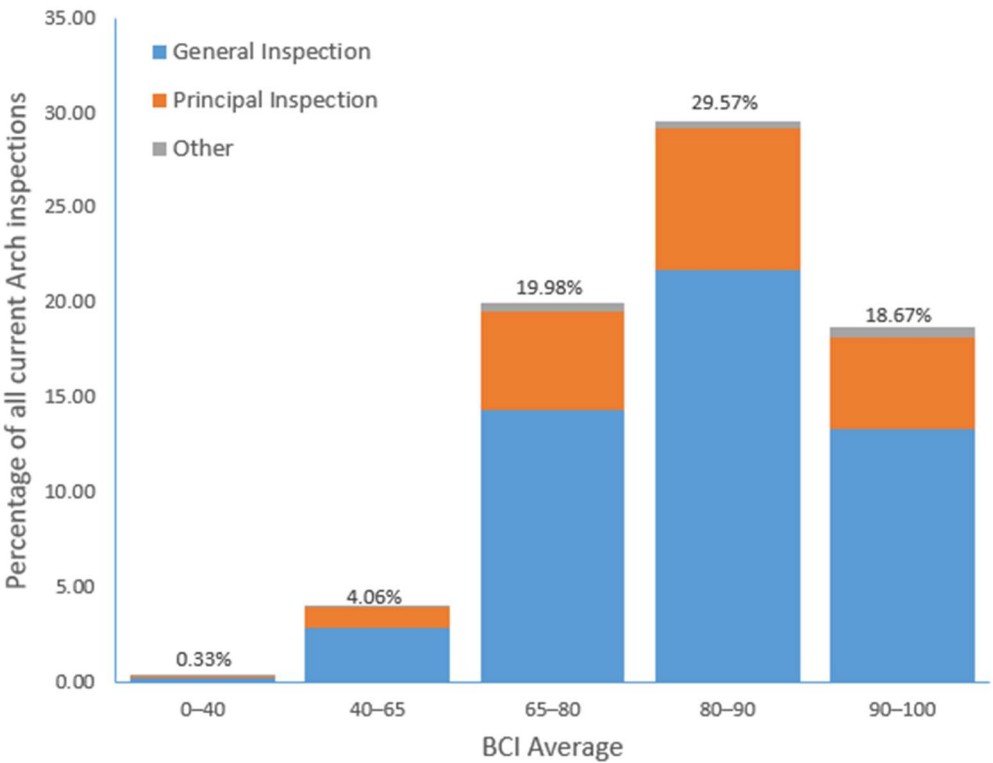

**Figure 1.** Grouped BCI average scores for all current arch bridge inspections in NI.

DfI bridges are separated into four different geographical divisional areas: the eastern 7.0% of all bridges and 6.2% of all arch bridges, northern 21.5% of all bridges and 26.1% of all arch bridges, southern 21.9% of all bridges and 23.6% of all arch bridges, and western 49.6% of all bridges and 44.7% of all arch bridges; see Figure 2. Each area is managed and inspected by separate divisional teams and consists of different total numbers of bridges.

The analysis of BCI scores for the four divisional areas provides a snapshot of the overall spread of arch bridge stock conditions in those geographical areas. The current BCI average inspection score for all arch bridge types in NI is plotted in ascending order for each geographical area in Figure 3, highlighting the overall spread of the condition in each area.

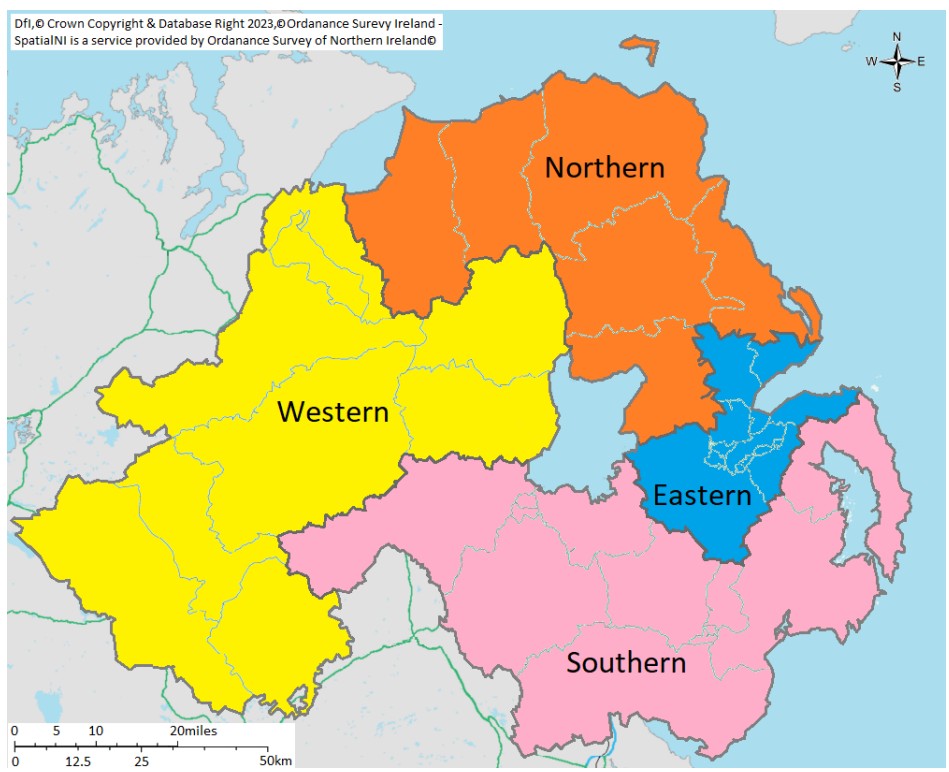

**Figure 2.** Four geographical divisional boundaries in NI (northern, eastern, southern, and western).

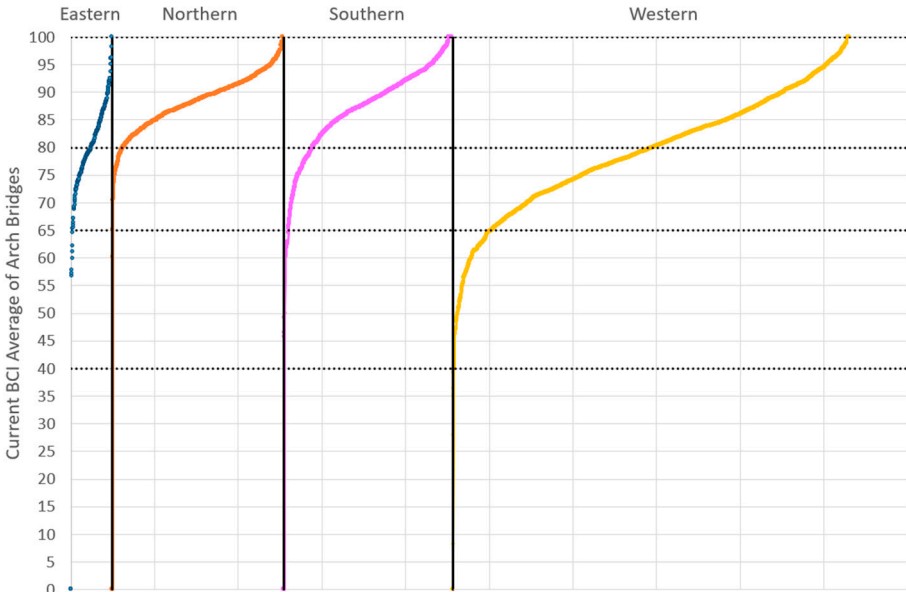

**Figure 3.** BCI Average scores for all current arch bridge inspections in NI are plotted in ascending order per divisional area.

It can be seen that western has more than double the number of bridges than the other divisions; however, the proportion of bridges with a BCI average value of 65 or below is greater than that of northern and eastern areas, which have only a small number of bridges in this zone. Figure 3 shows a snapshot of the regions with greater numbers of poor bridges and could be used as a starting point for allocating funding to maintenance spending in the future.

Given the distinct divisional divide and the fact that separate teams of inspectors cover these regional areas, the potential for engineering bias is considered a potential impacting

factor. Although bridge inspectors are trained to identify the typical defects, judging their extent and severity, ultimately, what one bridge inspector records could be subjectively different from another. It is possible for each individual or just one divisional team to be more risk-averse and rate certain defects higher than in another divisional area due to local knowledge or experience, as evidenced in previous research [3,4].

Currently, within DfI, inspectors undergo in-house inspection training; subsequently, line management undertakes a minimum of a 5% check on the inspections carried out by the bridge inspectors annually. The intention is that this should help with consistency, but as these checks are currently still within a divisional area, this regional variation could still have a bearing.

Plotting these BCI scores on a probabilistic distribution, as shown in Figure 4, indicates arch bridges in eastern and western divisions are in a poorer overall state than in the other areas, and northern bridges are in a better overall state.

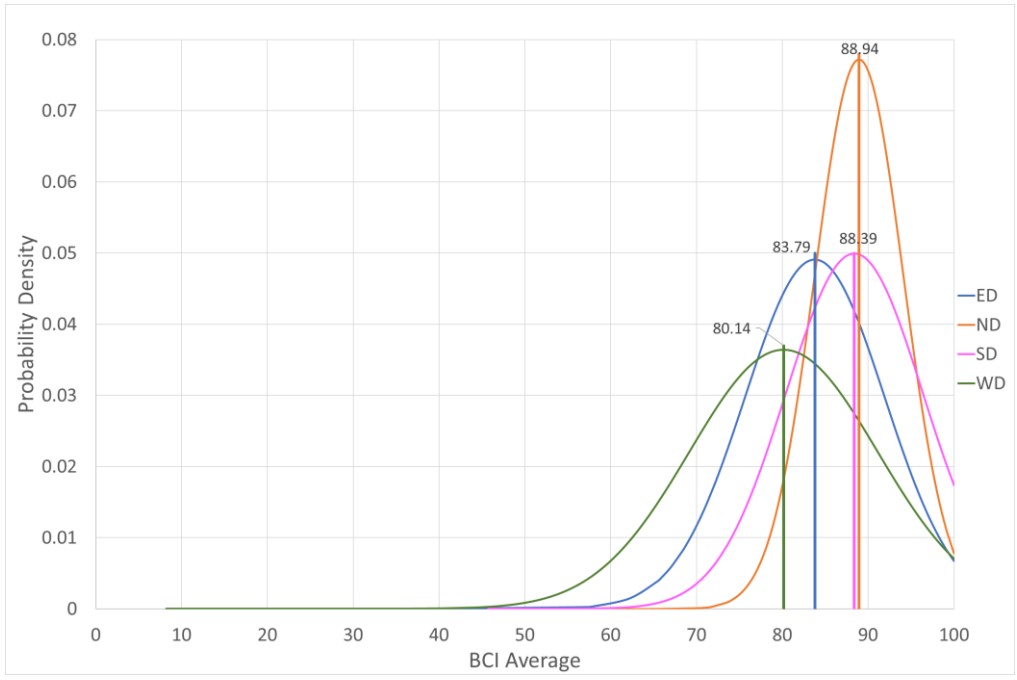

**Figure 4.** Probabilistic distribution of BCI scores by region. Eastern (ED), Northern (ND), Southern (SD) and Western (WD).

Western bridges had the lowest mean BCI average value score of approx. 80, with a shallower and broader peak on the normally distributed plot. This indicates a greater spread in the bridge conditions compared to the other divisions. Southern and eastern both have similar profiles; however, southern's mean BCI average score is approx. 88, which is 4 points higher than eastern's, which is 84. This indicates that both divisions have a similar spread of bridge conditions, but overall, the bridges are in slightly better condition in southern. Northern has a mean of approx. 89, which is comparable to the southern; however, the higher and narrower distribution than the other divisional areas indicates an overall better quality of bridge stock coalescing around the mean.

Given this observation alone, budget holders within the context of a finite budget could consider freezing or reducing the budget allocation for northern and reallocating it to western to try and bring these distributions closer to each other. Alternatively, they could increase the budget in the other divisions to bring them into a comparable overall condition. However, given the geographical consideration, it could be that the western has many more remote, lightly trafficked rural bridges, which are in a poorer but relatively steady state. Then if you consider eastern has larger more heavily trafficked urban bridges this alters the consideration of the maintenance priority. It could be that considering comparable

bridges, western bridges are comparable, but the picture is skewed by the relative increased numbers of rural bridges. Allocating funding based solely on the BCI output is therefore not fully appropriate and it is recommended that other factors such as a route importance factor should be considered during funding allocation. This highlights that other factors need to come into play to maximise the effectiveness of the maintenance programme and inform effective maintenance decisions.

## 2.2. Analysis beyond BCI to Identify Defect Trends in Arch Bridges

The next section of this paper presents an analysis of the 'augmented' data relating to approximately 20 years of condition and inspection data across 3549 arch bridges in NI before identifying a number of trends. Initial observations on this subset of bridge data for arch bridges indicate 94% of all arch bridges within this group are classed as 'roads over river', with an additional 2% are 'bridges over watercourses' (non-road carrying bridges such as pedestrian or cycle bridges over watercourses). Single-span bridges make up more than 95% of arch bridges and 99% are over 100 years old, 60% of bridges span up to 4 m, Figure 5.

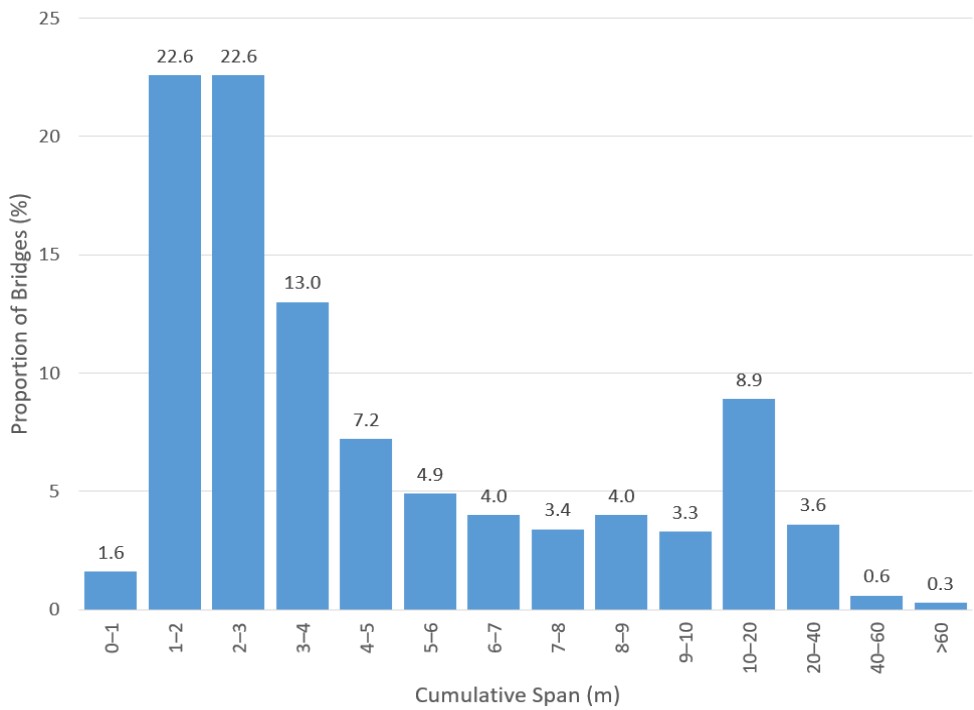

**Figure 5.** Distribution of cumulative spans for all arch bridges in NI.

This broadly follows the trend in railway bridges internationally, with [5,6] reporting that approximately 60% of the total bridge stock for railway bridges are arches, which represents around 220,000 bridges, with a majority of 85% being single span, 60% under 2 m or 80% under 5 m span, and approximately 70% being between 100 and 150 years old. [7] indicated that once you included national road networks, this would increase to around 300,000. However, when you consider that in NI alone there are approximately 3500 arch bridges on the road network, which is one of the smaller countries in the world, this figure is likely to be significantly higher.

Analysis of the specific defects recorded against arch bridges was carried out to try and identify trends in the deterioration of bridges in the intervening years. This was done to identify specific factors that may influence the deterioration of arch bridges in NI and ultimately compromise their remaining life if not adequately maintained. In the augmented dataset, a bridge component was a distinct part of the structure, as listed in Table 2. During a bridge inspection, a component is only recorded if a defect is present; therefore, the analysis of recorded components provides insight into the affected elements of the bridge.

**Table 2.** Distribution of defects for all arch bridges and overall priority/State 4 arch bridges.

| Component | All Defects (%) [1] | State 4 Defects (%) [1] |
|---|---|---|
| Parapet | 24.71 | 22.52 |
| Deck Soffit | 19.39 | 21.47 |
| Wingwall | 19.30 | 16.32 |
| Spandrel/Headwall | 14.22 | 13.53 |
| Abutment | 9.40 | 13.42 |
| Invert | 3.81 | 4.32 |
| Cutwater | 3.15 | 2.76 |
| Arch Ring | 2.60 | 2.03 |
| Pier Face/Column | 2.03 | 2.45 |
| Apron | 0.87 | 0.50 |
| Surface | 0.39 | 0.59 |
| Parapet Upstand | 0.08 | 0.07 |
| Abutment Slope | 0.03 | 0.03 |
| Movement Joint | 0.00 | 0.00 |

[1] Percentages rounded to 2 decimal places.

A further subset of this dataset was considered by filtering those bridges that were deemed an overall priority/State 4 to establish if any trends exist in these bridges compared to the rest.

A risk-based analysis that focuses on the component frequency for State 4 bridges identifies that the trend remained predominantly the same as that for all arch bridge defects, with Parapets being the most prevalent. The relative percentages have changed with marginal increases in the deck soffit, abutments, invert, pier face/column and surface percentage; see Table 2. The relative percentages of the parapet, wing walls, spandrel/headwalls, cutwaters, arch ring, apron, and parapet upstand decreased marginally. Abutments, piers, or columns and inverts support the deck soffit, so it follows that an increase in defects in one is likely to lead to increased defects in the other. Similarly, defects to the surfacing allow water into the deck and could translate into deck sofit defects also.

Parapets are one of the most visible parts of the structure and tend to be more exposed to the elements, vehicular traffic, and the associated deteriorative effects.

The inventory information held on these bridges highlighted that these bridges are predominantly single-span over rivers, and over 70% are located on C or U-Class roads on the rural network. All non-motorway roads in NI are designated in one of four categories: A, B, C and U, in order of significance to the network. The C- and U-class roads are typically minor roads within towns and most rural roads in the countryside. These roads often have sub-standard widths and geometry, which increases the effects of spray from passing traffic as well as the potential for collision/impact damage from agricultural and other traffic movements. These roads typically have grass verges and hedges, which encourage damaging vegetation growth with invasive root systems from large bushes, trees, or ivy if not controlled.

*2.3. Trends in Occurrence of Defect Types*

Analysis of the composition of defects logged for all the arch bridges and State 4 bridges demonstrates vegetation is the most prevalent in the inspection records, at 26.7% and 20.9%, respectively (Figure 6).

Bridges over rivers can provide favourable conditions for plant growth to thrive, with the structural elements of the bridge often providing support and shelter to young plants and gaps in the masonry allowing vegetation to take hold and exploit. The next most prevalent defects are pointing missing at approx. 15.3% (all arch bridges) and 12.1% (State 4), and then cracks at approx. 11.3% (all arch bridges) and 12.0% (State 4) (Figure 6). These defects are typically related to each other during inspections, as they follow the effects of vegetation damage by invasive root systems. Plants, such as tree saplings or ivy, can establish in very small gaps or joints in structures and with time, roots can expand and

exploit these gaps and loosen points and stones. This in turn provides more opportunities for plants to grow and if unchecked or repaired, these can induce cracks in a structure and cause significant damage over time.

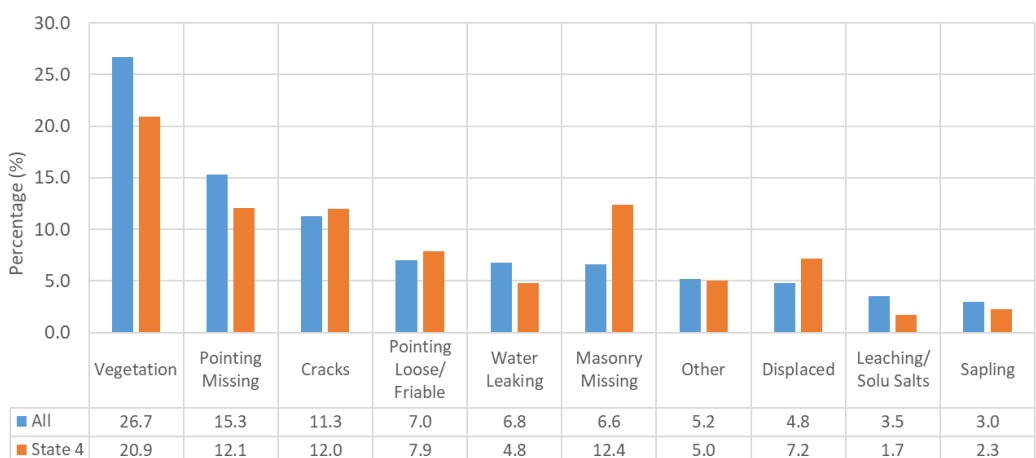

| | Vegetation | Pointing Missing | Cracks | Pointing Loose/ Friable | Water Leaking | Masonry Missing | Other | Displaced | Leaching/ Solu Salts | Sapling |
|---|---|---|---|---|---|---|---|---|---|---|
| All | 26.7 | 15.3 | 11.3 | 7.0 | 6.8 | 6.6 | 5.2 | 4.8 | 3.5 | 3.0 |
| State 4 | 20.9 | 12.1 | 12.0 | 7.9 | 4.8 | 12.4 | 5.0 | 7.2 | 1.7 | 2.3 |

**Figure 6.** Comparison between defect prevalence for all arch bridges (All) and State 4 arch bridges.

The data for the State 4 bridges follows the distribution of defects in a similar way, with a few notable exceptions. As arch bridges reach a poorer condition, the proportion of masonry missing has doubled from 6.6% to 12.4% (Figure 6). The relative proportions of vegetation defects, although still the most prevalent, have reduced from 26.7% to 20.9% and pointing missing from 15.3% to 12.1% in State 4 bridges (Figure 6).

The proportion of undermining and scouring has more than doubled, from 1.3% to 4.3% and 0.8% to 1.9%, respectively (Figure 7), when considering State 4 bridges.

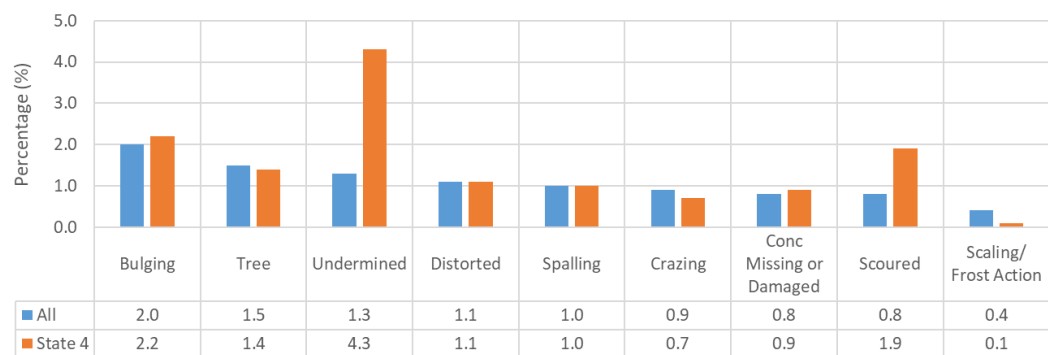

| | Bulging | Tree | Undermined | Distorted | Spalling | Crazing | Conc Missing or Damaged | Scoured | Scaling/ Frost Action |
|---|---|---|---|---|---|---|---|---|---|
| All | 2.0 | 1.5 | 1.3 | 1.1 | 1.0 | 0.9 | 0.8 | 0.8 | 0.4 |
| State 4 | 2.2 | 1.4 | 4.3 | 1.1 | 1.0 | 0.7 | 0.9 | 1.9 | 0.1 |

**Figure 7.** Comparison between defect prevalence for all arch bridges (All) and State 4 arch bridges, with adjusted Y-axis for clarity.

This demonstrates that these defects are more prevalent and may have a greater influence on the overall priority than the vegetation and pointing out missing defects alone.

### 2.4. Linking Component and Defect Types in Arch

Having examined the types of defects present in arch bridges, this section examines the types of defects present at the component level. When considering the defects for each component, vegetation is present in the most exposed components, followed by pointing missing and missing masonry. Figures 8 and 9 shows typical defect photos at two example masonry arch bridges.

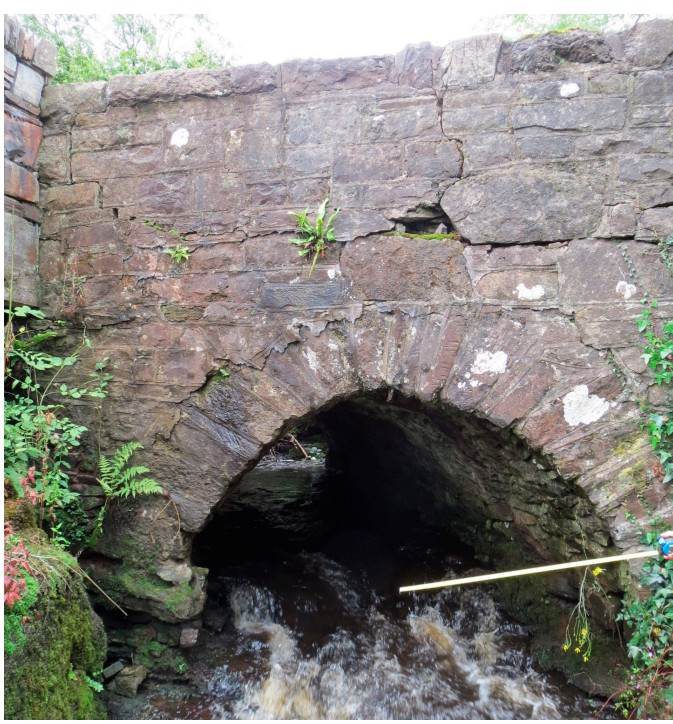

**Figure 8.** Typical defect photo shows scour to abutments, missing masonry, pointing loss, and cracking.

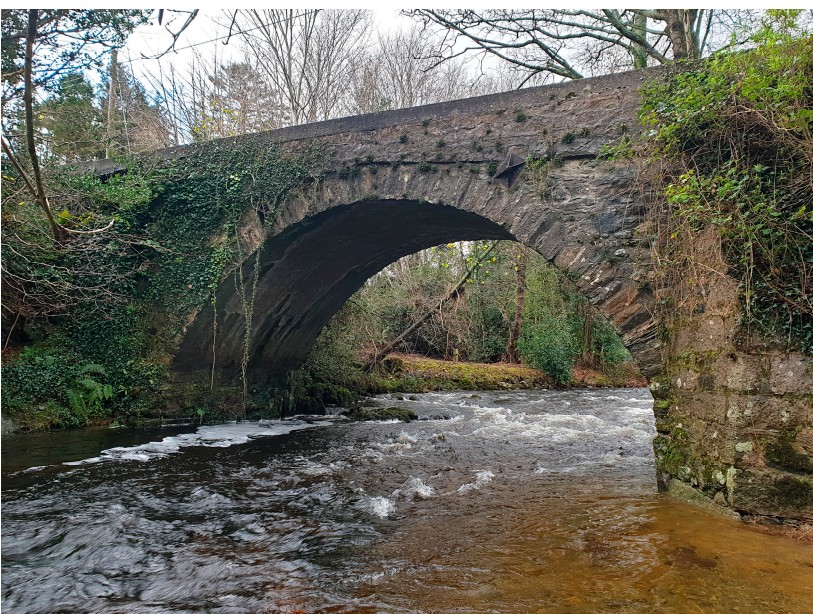

**Figure 9.** Typical defect photo shows scour to abutments and vegetation cover including trees.

The following section considers the change in the percentage that defects occur in the inspection record for all masonry arch bridge inspections versus the percentage that occurs in the State 4 arch bridges. This value is presented in Figure 10 to show the relative increase or reduction. Negative values represent an increase in the relative percentage of State 4 arch bridges, and a positive figure represents a decrease in the relative percentages of State 4 arch bridges.

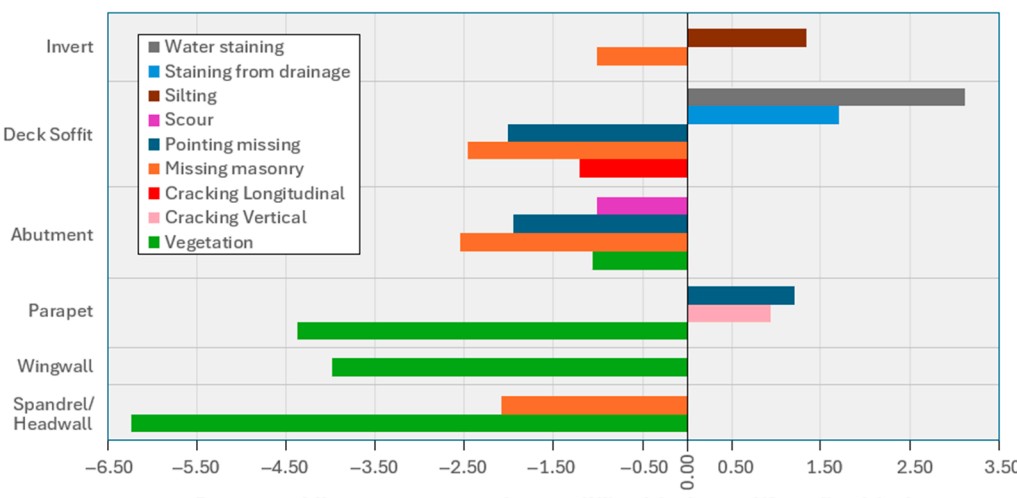

**Figure 10.** Comparison of component defect breakdown for all arch bridges (All) and State 4 arch bridges for selected defects and bridge elements.

As seen in Table 2, overall abutment defects increased in State 4 bridges, and looking at selected defects in Figure 10, 'Missing masonry', 'Pointing missing', 'Vegetation', and 'Scour' defects have increased.

Analysis of the defects associated with the Parapets identified a reduction in records overall for the State 4 bridges; however, 'Vegetation' increased while 'Pointing missing' and 'Cracking Vertical' decreased. This aligns with DfI treating parapet damage as a priority repair, as mentioned earlier in this paper, so it appears more frequently in the state 4 bridges and is repaired. Invert defects increased in State 4 bridges, with 'Missing masonry' increasing while 'Silting' decreasing.

The deck soffit defects are more prevalent in State 4 bridges with an increased proportion of 'Missing masonry', 'Pointing missing', and 'Cracking Longitudinal' noted while 'Water staining' and 'Staining from drainage' defects were reduced. Spandrel/Headwalls reduced in State 4 bridges but with 'Vegetation' 'Missing masonry' increasing. Wingwall defects reduced in State 4 bridges with "Vegetation' increasing.

It is worth noting that for the worst-conditioned State 4 bridges, the increase in 'Abutment', 'Deck soffit', 'Spandrel/headwall', and 'Invert' defects coincides with a greater proportion of 'Missing masonry'. Abutment also had an increase in 'Pointing missing' and 'Scour' defects.

This is logical, as when a structure is scoured or undermined, this removes the support for these components, resulting in the inability of the structures to effectively transfer loading through the deck into the support for the foundations. This can then lead to masonry dislodging in the abutments and deck soffit, increasing the 'Missing masonry' defects, which would result in additional movement. This encourages cracks, movement, and masonry to loosen and fall out, hence the increase in 'Displaced', 'Cracks', 'Pointing Loose/Friable', and 'Bulging' defects seen in Figures 6 and 7. Vegetation can also further exasperate this by exploiting these defects with invasive and destructive root systems. The culmination of these defects therefore compromises the structural integrity of the bridge asset, and so the risk to the structure is higher. This provides validation for the notion of critical elements and associated defects in these structures. This highlights the importance of understanding the hydraulic and geological properties of these sites to determine the impact they have on their structural behaviour.

Data associated with just the most recent current inspections for these arch bridges (data as of 23 April 2022) shows that the defect prevalence continues to follow this general trend presented in Figures 6 and 7. Vegetation represents 32.9% of all current defects on the arch, followed by 18.7% 'Pointing missing', 9.6% 'Masonry missing', and 8.9% 'cracks'. Similarly, it can be shown that the elements most reported follow a similar overall

general trend as that presented in Table 2. The elements most reported are in relation to the parapets 21.2% then primary deck elements (Soffit) 19.8% spandrel wall/headwalls 15.1% and wingwalls 15%. Followed closely by cutwaters, inverts, pier/column, and aprons. This review of the bridge condition data has provided insights into the prevalence of certain defects and the general condition of arch bridges within the network. However, more robust statistical methods are required to identify factors that impact the remaining life estimation of arch bridges.

*2.5. ANOVA to Identify Controls in the Deterioration of Arch Bridges*

Analysis of Variance (ANOVA) was introduced as a way of examining the impact of both bridge and environmental factors on the deterioration of bridges [8,9]. The use of ANOVA comes with several assumptions that need to be satisfied. These include the normal distribution of data and the homogeneity of variances across the data. In cases where these assumptions are not met, alternative tests can be used. In cases where the normality assumption is violated but there is homogeneity of variances, the Kruskal–Wallis test can be used. In cases where both assumptions are violated, Welch's trimmed test can be used.

ANOVA is used to investigate various factors to determine if they have a significant impact on the condition. In effect, it looks at the differences between the means of variations in datasets [10]. To conduct the ANOVA analysis on the deterioration of arch bridges, bridges that have no history of maintenance and a biennial inspection routine were used. A bridge with no history of maintenance must, by its nature, deteriorate over time and progress from its initial condition to a lower condition without intervention. As a detailed record of maintenance work within the dataset held is not always fully populated, it is necessary to filter those showing an improvement in their condition. Where an improvement in the bridge condition, for example, from condition State 3 to condition State 1, is identified, these must be removed from the analysis as this improvement could not have been possible without intervention taking place. This assumption is necessary as an improvement in the bridge condition could only indicate maintenance, and these data points need to be removed due to a lack of data to prove otherwise. Similarly, the ability to assess the deterioration to State 4 is also challenging. When bridges reach State 4, the level of deterioration is such that they are generally either replaced, repaired, or abandoned, so an analysis of this state is not possible.

To conduct this test, each state is taken in turn, and the next conditional state reached was determined. The null hypothesis, $H_0$, states that for bridges in a certain condition state, the mean values, $\mu$, of the next condition state are equal among the categories of the factor being investigated [11].

$$H_0: \mu_1 = \mu_2 = \mu_3, \tag{1}$$

The alternative hypothesis, $H_a$, states that the mean values, $\mu$, are not equal.

$$H_a: \mu_1 \neq \mu_2 \neq \mu_3, \tag{2}$$

Then check the 95% confidence level, $F$, by considering the Sum of Squares between groups of data, SSb, and within the group SSw, and Degrees of Freedom between groups of data, Dfb, and within the group Dfw. Then consider the Mean Squares between groups of data, MSb and within the group MSw.

$$F = \frac{\left[\frac{SSb}{Dfb}\right]}{\left[\frac{SSw}{Sfw}\right]} = \frac{MSb}{Msw} \tag{3}$$

If the test is significant (i.e., 5% level, *p*-value), then there is sufficient evidence to reject the null hypothesis in favour of the alternative. This means that the factor has a significant impact on the deterioration at that stage in its condition. The *p*-value corresponds to the

remaining area under the distribution curve outside that is covered by the 95% confidence level. Firstly, the influence of the number of spans was considered, as was whether it was single or multi-span. This analysis indicates that during the initial stages of deterioration, this does have a significant impact; however, this declines as the structures' condition reaches State 3; see Table 3. When considering the 10% confidence level, the decline in State 3 also becomes significant.

**Table 3.** ANOVA analysis: Summary.

| Feature | State | Significance at 5% Level |
|---|---|---|
| Single Span versus Multi-span | 1 | Significant |
| | 2 | Significant |
| | 3 | Not Significant (Significant at 10% Level) |
| Road Over River versus Not Over River | 1 | Not Significant (significant at 10% Level) |
| | 2 | Significant |
| | 3 | Not Significant |
| Road Over Water versus Not Over Water | 1 | Not Significant |
| | 2 | Significant |
| | 3 | Significant |
| Deck Widths | 1 | Not Significant |
| | 2 | Significant |
| | 3 | Not Significant |

Secondly, the influence of the bridge function and whether it was a 'road over river' or 'road not over a river', were tested. During the initial and late stages of deterioration, this appears to not have a significant bearing on the change in state; however, at the 10% confidence level, it does appear to become significant in the initial state. When moving from State 2, it did indicate a significant effect; see Table 3.

The influence of whether the bridge was 'road over water' or 'road not over water' indicated a significant impact on later stages of deterioration but not in the initial stage.

It is worth noting that 'road over river' relates to road bridges over rivers. Rivers tend to be flowing bodies of water, which have the potential for transporting debris and therefore associated damaging impact effects and scouring. 'Road over water' includes bridges over canals, lakes, and ponds, which are largely stationary bodies of water with less potential or moving debris and scouring. As such, this can skew the results and could partially account for the difference in the analysis in Table 3.

For the case of deck width, this continuous variable was broken up into five categories. The analysis indicated that the deck width has a significant impact on the core stages of the bridge's lifespan and is not significant in the early and later stages of deterioration.

In summary, these results show what factors have an impact on the deterioration of masonry arch bridges in NI. In the initial stages of the deterioration of the bridge, the number of spans is the most significant. However, the table also shows that if the *p*-value is considered at the 10% level, then 'road over river' becomes a significant factor. This is not true for 'road over water' as this would also include bridges over slow-flowing bodies of water, such as canals. This indicates that, rather than just the presence of water, river flow has an impact on the deterioration of arch bridges. Therefore, this should be considered when predicting the estimated remaining life of arch bridges.

When considering the deterioration beyond condition 3 the 'road over river' is not significant. Although this seam incongruous with the earlier states, the absence of bridge data in this category explains this result. As mentioned earlier in this paper, arch bridges would generally be maintained, repaired, replaced, or decommissioned before deteriorating to this condition.

This provides a means of developing an additional prioritisation tool to identify which arch bridges within the network are likely to need maintenance first.

This further supports the selection criteria, with the general condition of bridges across the network, the predominance of defects, and recorded components, as detailed in previous sections.

## 3. Conclusions

This paper presents the background to the dataset held by DfI within the NI bridge management database. An assessment of the current overall bridge stock condition BCI scores for NI identified trends in regional differences, which indicates that bridges in the western division had a lower average BCI and greater spread in condition than the other divisions, and it was postulated that this may be due to geographical features or human factors.

As arch bridges constitute 53% of the total bridge stock and have inherent concerns regarding the uncertainty around material type, construction properties, design life, and load capacity, these bridges were selected for further investigation.

This research has presented the initial data analysis for arch bridges, the prevalent defective components and defect types, and how they interact and change with deteriorating overall priority. The results highlight vegetation as the most prevalent defect noted and parapets as the most reported component of all the bridge conditions. When considering only the State 4 arch bridges, scouring and undermining proved to be significant factors. These bridges tend to be older structures with typically unknown foundation depths or conditions, which poses a significant management problem for DfI. The ANOVA identified function, number of spans, and deck width as significant factors in deterioration at various stages in a bridge lifecycle. In the initial stages of the bridge condition, the number of spans was significant; in the middle stages, the number of spans was significant, whether it was over a river or water; and the deck width proved significant, whether the bridge was over water or not.

Although this review has considered a range of factors, these are limited to masonry arch structures and do not attempt to account for the bearing human influence has on the overall dataset. Conclusions have been made based on a dataset that the author has acknowledged could be influenced by geographic or human factors. No attempt was made in the analysis to account for this potential bias or normalize the results based on individual inspectors or regions. It is likely to prove difficult to undertake such a task, although it is still appropriate to use the results in this paper to identify general trends in the data.

The findings in this paper and further analysis of this dataset have been used to review the management of structures within DfI, which highlighted the need for specific bridge inspection training. A review questionnaire has been issued to bridge inspectors to complete individually to try and establish if bias exists. A new training course is being developed with the assessment to ensure consistency in approach as well as focusing on elements of concern that have been identified in this report. The course will focus on the effects of various defects, such as those caused by vegetation and scour/undermining on masonry arch structures, and ways to identify them during inspections. This research has also confirmed the need for a more proactive approach to scour management and has resulted in the development of a real-time scour sensor to complement the inspection process.

**Author Contributions:** K.C. wrote the paper including Introduction, K.C.; Analysis of the current Bridge Management System Data, K.C.; Network-wide BCI distribution, Analysis beyond BCI to identify defect trends in arch bridges, K.C.; Trends in Occurrence of Defect Types, K.C.; Linking Component and Defect types in the arch, K.C.; ANOVA to identify controls in the deterioration of arch bridges, K.C. and N.-A.S.; Conclusions, K.C.; Acknowledgements, K.C. N.-A.S. provided the data presented in the ANOVA to identify controls in the deterioration of arch analysis section M.L. and S.T. provided review and comments during the writing process. All authors have read and agreed to the published version of the manuscript.

**Funding:** This research was funded by the Department for Infrastructure (DfI) with financial support from the Royal Academy of Engineering under the research fellowship program (RF_201718_1796).

**Data Availability Statement:** The data presented in this study are available on request from the corresponding author. The data are not publicly available due to the nature of the information and are at the sole discretion of the Department for Infrastructure Northern Ireland.

**Acknowledgments:** The authors would like to thank the Department for Infrastructure (DfI) for access to the complete bridge management records, technical and financial support, and for allowing the analysis and findings to be used in this paper. The financial support of the Royal Academy of Engineering under the research fellowship program is also gratefully acknowledged.

**Conflicts of Interest:** The lead Author is an employee of the Department for Infrastructure Northern Ireland, which may be perceived as inappropriately influencing the representation or interpretation of reported research results.

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
