# Peer review of "Analysis of Arch Bridge Condition Data to Identify Network-Wide Controls and Trends"

_infrastructures, doi:10.3390/infrastructures9040070_

Round 1

Reviewer 1 Report

Comments and Suggestions for Authors

Drawing on the historical data examining arch bridge conditions in Northern Ireland, the authors have pinpointed patterns and causes of bridge deterioration and determined what factors impact the deterioration of the arch bridges. The study asserts that recognizing these patterns will assist relevant bodies in prioritizing and effectively maintaining bridges.

The article offers strong arguments and demonstrates how data that may initially appear outdated can be repurposed effectively. Although the paper's content is commendable, it would benefit from revisions to enhance its quality and appeal to readers.

This document contains reviews of the paper, organized in a manner that starts with the major issues and then proceeds to address the minor ones.

1.       Figures 8 and 9 on page 10 contain crucial data but require presentation improvements for better clarity. Currently, the range of colors used to depict various types/causes of deterioration is indistinct. The use of similar color palettes makes it challenging to distinguish between them visually. The section from lines 260 to 276 on page 11 could be significantly improved in its presentation. As it stands, the structure is repetitive and lacks coherence. This portion of the text pertains to Figures 8 and 9 on page 10. Given that Figures 8 and 9 are difficult to interpret, the information in this paragraph is challenging for readers to verify and understand.

2.       Figure 3 on page 5 requires more explanation. Why does the data spread that represents different geographical regions are of different sizes? What is the significance of such representation? Furthermore, the labeling at the top of the figure is inconsistent, with text not arranged uniformly. The labels for “Eastern”, “Northern”, and “Southern” are on the same level, whereas “Western” is positioned at a different level.

3.       In section 2.5 on page 12 of the manuscript, incorporating a mathematical representation of the Analysis of Variance (ANOVA) would enhance comprehension for the readers.

4.       In the manuscript's conclusion, spanning pages 13 and 14, from lines 379 to 399, the author summarizes the research findings. Nevertheless, this section does not sufficiently address the implications of these findings. Also, addressing the study's limitations could enhance the conclusion's quality. Additionally, it is recommended to propose potential future research directions building upon the current results.

5.       The legends GI, PI, and Other in Figure 1 on page 3 require further explanation. It’s unclear what these terms represent and this needs to be clarified.

6.       The term “p-value” mentioned in line 363 on page 13 needs additional explanation to ensure that readers have a clearer understanding of its meaning.

7.       There is a typographical error in line 273 on page 11. The word “deceased” should be corrected to “decreased”.

Author Response

Please see attachment for word doc detailing changes made which I hope addresses the comments. Thank you. 

Also summarized below

1- Agreed, new figures have been presented reducing the info presented to make it clearer to interpret. These figures attempted to display too much information at once, I have attempted to simplify the figures only presenting the relative difference in occurrence of the defects between All and State 4 now for specific defects and bridge elements and reduced the discussion to match. I hope this clarifies the Reviewers concerns.

2- The different spreads are due to the different areas having different numbers of bridges. Western has allot more bridges than Eastern so a widder range. This figure really is to show the BCI values in ascending order relative to each divisional area. Text amended to reflect.

3- Noted, added equations to aid comprehension as suggested.

4- Noted, I have included a section covering the limitation regarding not addressing the potential human bias. Also included future work, where this research was used to inform a review of the management of structures in DfI and led to bridge inspector training focusing on vegetation and scouring around arch bridges. I also note that this research was used to justify the development of a scour sensor which will be the subject of subsequent papers. 

5- A paragraph has been included to explain these terms prior to the figure and GI PI are not used in the revised legend.

6- Equations added to aid comprehension also act to clarify ‘p-value’.

7- Noted and changed

Reviewer 2 Report

Comments and Suggestions for Authors

 It is recommended to attach photographs of typical damage conditions for different types of Bridges.

Comments on the Quality of English Language

There are some mistakes on spelling and punctuation.

Author Response

Noted, although to provide photos for all defects would be problematic with the limitations on the length of the journal. I have included two if deemed useful.

I have also addressed some grammatical mistakes.

Thank you

Please see revised paper with track changes showing all changes made in response to all reviewers comments

Reviewer 3 Report

Comments and Suggestions for Authors

A review paper describes structural elements and associated defects recorded in the inspections of arch bridges through various databases from Northern Ireland (NI) by the Department for Infrastructure. The review paper provides valuable results and could be accepted after addressing following comments:

 -        The first paragraph of the introduction section is not clear. Pleas rewrite for more carity.

 -        Section 2: provide more background in regard to network wide BCI distribution; starting with NI bridge stock.

 -        Edit the Figure 1 to convey more academic style than a report. Please see journal requirement.

 -        Replace Figure 2 to a higher resolution Figure as it is hardly can be seen.

 -        References are not adequate for a review paper. A revised list will be required.

Comments on the Quality of English Language

A through proof-reading check is required. Some grammatical and spelling errors are found.

Author Response

1- Reworded to clarify,

2- Background to the BCI is in the previous section however a few sentences have been included in section 2 to try and address this comment.

3- Figure edited to use correct font and replicate figures published previously in this journal.

4- Noted and done.

5- Additional references added although this paper is not intended to be a review paper.

6- Corrections made.

I have included a track changes version of the paper to show all the changes I have made in response to all 3 reviewers for reference. I hope this addresses your concerns. I can upload a clean copy if you all agree. Thanks

Round 2

Reviewer 1 Report

Comments and Suggestions for Authors

The authors have addressed all comments from the reviewer.